# Effect of Steaming Processing on Phenolic Profiles and Cellular Antioxidant Activities of *Castanea mollissima*

**DOI:** 10.3390/molecules24040703

**Published:** 2019-02-15

**Authors:** Fangyuan Zhang, Fengyuan Liu, Arshad Mehmood Abbasi, Xiaoxiao Chang, Xinbo Guo

**Affiliations:** 1School of Life Sciences, Southwest University, Chongqing 400715, China; fyzhang@swu.edu.cn; 2School of Food Science and Engineering, South China University of Technology, Guangzhou 510640, China; phoyueen@outlook.com; 3Department of Environmental Sciences, COMSATS Institute of Information Technology, Abbottabad 22060, Pakistan; arshad799@yahoo.com; 4Institute of Fruit Tree Research, Guangdong Academy of Agricultural Sciences, Key laboratory of South Subtropical Fruit Biology and Genetic Resource Utilization (MOA), Guangdong Province Key Laboratory of Tropical and Subtropical Fruit Tree Research, Guangzhou 510640, China

**Keywords:** chestnut, steaming, phenolics, antioxidant activity, cellular accessibility

## Abstract

The intention of this study was to investigate the effect of steaming processing on phenolic profiles and antioxidant activities in chestnuts. Steaming processing at different times and temperatures depicted diverse impacts on free and bound fractions. Though, bound phenolics were stable but long time steaming at higher temperatures tended to improve the levels of phenolics, flavonoids as well as antioxidant activities in chestnut kernels, by up to 60.11% of the original value. Seven phenolic compounds including ferulic acid, chlorogenic acid, gallic acid, vanillic acid, syringate, *p*-coumaric acid and quercetin were found to change during thermal processes. Significant relationships (*p* < 0.05) were identified between total phenolics and total antioxidant activities. However, the consistency of chlorogenic acid (*p* < 0.01) only with cellular antioxidant activity indicated poor bio-accessibility of the phytochemicals in chestnuts. However, this situation could be partly improved by steaming. Steaming could improve the cellular accessibility of free phytochemicals, particularly, increasing the bio-accessibility by 41.96%. This study provided valuable information on dynamic changes of phenolic profiles and antioxidant activity of chestnuts under a steaming process, which could offer possible guidance for the chestnut processing industry in the future.

## 1. Introduction

Chestnut (Castanea) is a member of the Fagaceae family and mainly distributed in Europe, East Asia as well as North America. Chinese chestnut (*Castanea mollissima* Blume) is a native species in mainland China with a cultivation history of over 3000 years [1]. Chestnut has long been used as a kind of cosmetic supplement in Korea [2]. Moreover, chestnuts are a healthy food resource with high nutritional values, containing high levels of carbohydrates and proteins but zero cholesterol and gluten [3]. These features make chestnuts distinguished in the nut family and thus heartily accepted by people around the world [4]. 

Epidemiological studies indicated that sufficient daily intake of fruits and vegetables could lower the risk of cardiovascular diseases, cancers and diabetes [5]. Phenolics, as secondary metabolites in plants, play an important role in the prevention of these illnesses [6]. However, studies about phenolics in chestnut fruits are rare. The main reason for this might be the relatively low content of phenolics in chestnut fruit compared to its leaves, skins, flowers [7] and even other kinds of nuts [4]. Therefore, many researchers have turned their attention to other resources from chestnut trees, such as chestnut leaves, bark and so on [2,8]. Nevertheless, we believe that the phenolic content in chestnut fruit might be underestimated, because fresh-cut chestnuts turn brown very quickly due to enzymatic browning caused by polyphenol oxidase (PPO) and peroxidase (POD). Such activities lead to nutrition losses and quality degradation of chestnut fruits [9]. Proper treatments such as thermal processing could inactivate such enzymes, consequently protecting vulnerable nutrients as well as bioactive compounds [10]. In addition, phenolics usually bound by protein, cellulose or other nutrients in food matrices, also decrease their bioavailability. However, thermal processing could destroy these structures releasing bound phenolics and increasing the phenolic contents [11].

Chestnuts are rarely consumed raw. However, limited literature has reported the effect of processing parameters on phenolic compounds in chestnut fruits. Zhu summarized that processing time (roasting and boiling) might influence phenolic content in chestnut fruits [12]. Hou et al. reported that frying decreased total phenolics and flavonoids in chestnut fruits [13]. Meanwhile, according to Li’s report, roasting had no significant influence on total phenolic and flavonoid contents in chestnut fruits but frying and boiling slightly decreased their contents [14]. Several studies have reported the changes of total phenolics at different stages of industrial processing [15,16]. However, the effect of steaming on phenolics has rarely been studied so far in chestnut. Steaming is the best thermal processing method to preserve these phytochemicals in fruits and vegetables [17]. Such as in our previous study, compared to boiling and microwaving, the chestnut fruits processed by steaming had the highest phenolics and flavonoids [18]. While, the changes of phenolic profiles and antioxidant activities in chestnut with different treatments of thermal processes were less known in previous research.

Moreover, most of the studies mentioned above ignored bound phenolics in chestnut fruits. Although, changes in total phenolics after different types of processing have been reported previously [13,14,19]. However, such studies cannot reveal the changing mode behind these phenomena. Dewanto et al. stated that thermal processing could destroy food matrices and release bound phenolics [20]. In addition, researchers have reviewed the importance of protein-phenolic compound interactions in food [21,22]. Steaming and other thermal treatments could increase the amount of bound phenolics by chemical reactions, as the oxidations of *o*-diphenols to *o*-quinones give place to polymers and pseudo-melanin material causing the browning of the vegetal tissue [11]. Therefore, it is important to include the contents of bound phenolics in the processing study. Bound phenolics might be increased during steaming, because our previous study has indicated that steaming could dramatically increase both the bound and free phenolics in chestnut fruits [18]. Based on this finding, we hypothesized that the processing temperature and time would play an important role in this reaction. To better understand the changing patterns of phenolics during steaming, here the influence of steaming time and temperatures on both free and bound phenolic compounds as well as their antioxidant capacities in chestnut fruit were further investigated.

## 2. Results

### 2.1. Changes of Total Phenolics and Flavonoids after Processing

Changes of total phenolic contents (TPC) with different thermal processing conditions are shown in Figure 1. The data are expressed as milligram gallic acid equivalent per 100 gram in fresh weight (mg GAE/100 g FW). The content of total phenolics was 38.43 ± 0.68 mg GAE/100 g FW in fresh chestnut. While, the highest content of total phenolics increased up to 62.52 ± 1.19 mg GAE/100 g FW after thermal processing in chestnut fruits, which was 1.63 folds higher than the initial content. Free phenolic content of chestnut was significantly increased with the time-kinetics manner at the same temperature (100 °C) treatment, and up to 55.05 ± 1.13 mg GAE/100 g FW after 30 min processing, which was 1.54 folds of the initial content (Figure 1a). While, the content of free phenolic slowly increased with the thermal gradient manner with the same processing time (20 min), and up to 53.72 ± 2.17 mg GAE/100 g FW at 121 °C treatment (Figure 1b). Bound phenolic content was dramatically increased in chestnut with a thermal gradient manner as well as time-kinetics manner, and the highest content was 8.62 ± 4.24 mg GAE/100 g FW at 121 °C for 20 min processing, which was 3.14 folds of the initial content (Figure 1b). Even the bound phenolics significantly increased in the chestnut during thermal processing; it contributed little to total phenolics (less than 13%). While free phenolics played the main contribution to total phenolics in chestnut, about 90% did during thermal processing.

Changes to the total flavonoid contents (TFC) with different thermal processing conditions are shown in Figure 2. The data are expressed as milligram catechin equivalent per 100 gram in fresh weight (mg CE/100 g FW). The total content of flavonoid was 53.16 ± 1.04 mg CE/100 g FW in fresh chestnut, and 43.59 ± 0.40 mg CE/100 g FW in free fraction and 9.57 ± 1.12 mg CE/100 g FW in bound fraction, respectively. After steaming processing, total flavonoid content dramatically increased up to 71.51 ± 0.62 mg CE/100 g FW in chestnut, which was 1.35 folds of the initial content. Free and bound fractions increased the highest up to 1.22 and 2.13 folds as compared with the initial content, respectively. Moreover, the rate of bound to free fractions increased from 21.95% to 52% after the thermal processing. The changes of flavonoid in chestnut kept the same trends with the changes of phenolics during thermal processes.

### 2.2. Changes of Phenolic Components after Processing

In total, seven phenolic compounds including ferulic acid, chlorogenic acid, gallic acid, vanillic acid, syringate, *p*-coumaric acid and quercetin were identified and quantified in free and bound fractions of chestnut fruit by reverse phase-high performance liquid chromatography (RP-HPLC) as shown in Table 1. Chlorogenic acid was the dominant phenolic compound in free fractions of chestnut and below the detected limit in bound fractions. In free fractions, the content of chlorogenic acid varied from 15.16 ± 0.13 to 22.42 ± 0.01 mg/100 g FW with time-kinetics manner during 100 °C steaming processing, about 1.2 folds increased as compared with the initial content. Gallic acid was the second abundant phenolic compound, and it increased up to the highest of 12.93 ± 0.06 mg/100 g FW (100 °C, 20 min) and declined up to the lowest of 4.99 ± 0.11 mg/100 g FW (115 °C, 20 min) in free fraction, which was 1.33 and 0.51 times the initial content, respectively. While gallic acid declined from 5.55 ± 0.01 to 2.86 ± 0.01 mg/100 g FW in bound fraction with time-kinetics manner at 100 °C steaming processing, and it increased with thermal gradient manner from 100 °C and recovered to the initial content at 121 °C processing. The content of free ferulic acid was inconspicuous changed during thermal processing, while the bound ferulic acid varied from 2.92 ± 0.43 to 4.16 ± 0.05 mg/100 g FW with time-kinetics manner. p-Coumaric acid was kept consistent in free and bound fractions during thermal processing. Vanillic acid and syringate increased a little in free fraction during thermal processing, and they were below the detection limit in bound fraction. The changes of quercetin in free fraction significantly declined during thermal processing, while it dramatically increased in bound fraction after thermal processing. Overall, increasing patterns were observed in the majority of the studied phenolic compounds with different steaming processing treatments. However, a decrease in free ferulic acid, bound gallic acid, free syringate and free quercetin was also observed after processing.

### 2.3. Changes of Total Antioxidant Activities after Processing 

Total antioxidant activities of chestnut were analyzed by oxygen radical capacity (ORAC) assay and presented as micromole Trolox equivalent per gram in fresh weight (μmol TE/g FW) in Figure 3. Total antioxidant activity was 8.32 ± 1.28 μmol TE/g FW in fresh chestnut, and 6.36 ± 1.13 μmol TE/g FW in free fraction as well as 1.97 ± 0.10 μmol TE/g FW in bound fraction, respectively. Total antioxidant activity dramatically increased up to 13.36 ± 1.94 μmol TE/g FW after thermal processing, which was 1.61 folds of the initial activity. Antioxidant activities in free fraction significantly increased with the manner of time-kinetics and thermal gradients, and up to the highest activity of 10.46 ± 1.90 μmol TE/g FW (121 °C, 20 min), which was a 1.65 fold increase as compared with the initial. While, the bound antioxidant activities experienced little changes during thermal processing.

### 2.4. Changes of Cellular Antioxidant Activities after Processing

Cellular antioxidant activity (CAA) of chestnut was studied by CAA assay and the results are presented as nanomole quercetin equivalent per gram in fresh weight (nmol QE/g FW) in Table 2. The CAA assay simulates some of the cellular processes including bio-accessibility, cell uptake, metabolism, and distribution of bioactive compounds to predict the antioxidant behavior in biological systems [23]. PBS and No-PBS wash treatments were run in this assay to evaluate intracellular and extracellular antioxidant activities of extracts and also to analyze the cellular uptake of extracts. Comparatively, CAA was higher in free fractions than bounds. In free fraction, the CAA value of the No-PBS wash increased up to the highest of 2.27 ± 0.10 nmol QE/g FW (100 °C, 40 min) with the time-kinetics manner, and the value of PBS wash also increased up to 0.45 ± 0.06 nmol QE/g FW (100 °C, 30 min). While thermal gradients, little affected the cellular antioxidant activities in chestnut. In bound fraction, the cellular antioxidant activities of PBS and No-PBS wash treatments also changed a little with thermal processing in chestnut. While the rate of cellular uptake (the rate of PBS value with No-PBS value) significantly increased during thermal processing in free fraction.

### 2.5. Correlation Analysis 

Results showing association of phenolic composition, total and cellular antioxidant activities as well as steaming processing in chestnut fruits are given in Table 3. Total flavonoid contents in chestnut fruit were strongly correlated with total phenolics (*p* < 0.01). However, correlation between flavonoids and ORAC at *p* < 0.05 indicates that in chestnut fruit, flavonoids had little contribution to the antioxidant activity in vitro. In addition, total ferulic acid, total chlorogenic acid as well as total vanillic acid exhibited significant correlation (*p* < 0.01) to total phenolic contents (vanillic acid also associated strongly with total flavonoid contents with *p* < 0.01). Furthermore, chlorogenic acid, which was the dominant phenolic, depicted a significant relationship with ORAC. 

Conversely, high levels of phenolic contents would not necessarily associate with high CAA values. As shown in Table 3, total phenolics and flavonoids contents, ORAC values as well as individual phenolic compounds didn’t show relative correlation with total CAA values. Only, chlorogenic acid showed strong association (*p* < 0.01) with the total CAA values in the case of no PBS wash samples but did not well correlate with the total CAA values in PBS wash samples. This suggested that the cellular accessibility of chestnut extracts was low, which was consistent with the results from Table 2. While, *p*-coumaric acid was found to have a significantly negative correlation with total and cellular antioxidant activities. This indicated that *p*-coumaric acid contributed less to antioxidant properties in chestnut fruits.

## 3. Discussion

### 3.1. Influence on Phenolics and Flavonoids

The present study revealed that during steam processing, both time and temperatures could have positive and negative effects on phenolic profiles as well as ORAC values in chestnut fruits. Although, steaming at 100 and 121 °C for 40 and 20 min, respectively significantly improved the levels of phenolics, flavonoids and ORAC values compared to other groups. Moreover, steaming at 100 °C for 20 min and 115 °C for 20 min decreased the levels of flavonoids. This indicates that even with the same processing method, variation in time and temperature could have different impacts on phytochemical composition and chemical antioxidant activity. Likewise, Goncalves et al. [24] reported that roasting at 200 °C for 40 min increased the total phenolic contents in the chestnut fruit. However, Nazzaro et al. [25] discovered that roasting observed a decrease in total polyphenol content and antioxidant activity after 200 °C for 20 min. It has been reported that degradation of oxidases, breakage of phenolics from food matrices and synthesis of novel phenolics might be involved in the enhancement of phenolic compounds [11,18], while a decrease in phenolics may be due to the reduction of free phenolic profiles (Figure 1). Additionally, bound phenolics were more stable under a different processing time and temperatures. Our study revealed steaming for a longer steaming time at a high temperature led to higher phenolic contents and ORAC values. This confirmed the assumption about release or formation of phenolics in chestnut fruit during steaming, as steaming at 100 °C for 10 min is enough for disruption of oxidases [26]. Conversely, a decrease in flavonoids content was observed after steaming for 20 min at 100 and 115 °C, which might be due to the reduction in free flavonoids. Although it is rational to observe the decrease in flavonoids, which are more susceptible to heat than phenolics [17]. Our findings revealed that steaming at 100–115 °C for 20 min was the critical point to increase flavonoid levels in chestnut fruits where the release or formation of flavonoids couldn’t compensate the loss of flavonoids.

### 3.2. Revelation from Phytochemical Profiling by HPLC

Previous studies have pointed out that Gallic acid and ellagic acid are the dominant phenolics in European chestnut fruit [16,24]. In the present study, however, Gallic acid was higher in concentration but ellagic acid was below the limit of detection. This discrepancy might be due to the difference in chestnut varieties [19]. In addition, the level of bound phytochemicals in chestnut fruits was lower than free phytochemicals, and several bound compounds were below the detection limitation. This indicated that free phytochemicals played more of a contribution in the antioxidant activities of chestnut fruits. 

Interestingly, chlorogenic acid was the dominant phenolic compound in free fraction, but it was below the detection limitation in bound fraction including the fresh group (Table 1). In addition, free chlorogenic acid increased after steaming and was highest in samples steamed at 100 °C for 40 min. These findings were analogous to Hwang’s research, who reported that heat treatment could induce the formation of chlorogenic acid [27]. Likewise, Dawidowicz et al. also reported isomerization and transformation of chlorogenic acid during thermal processing at temperatures ranging from 100–200 °C [28]. Therefore, it could be speculated that similar reactions also took place in chestnut fruits during steaming. 

Similarly, the performance of ferulic acid contradicted previous reports in sweet corn [20]. Although the total ferulic acid increased after steaming, unlike the previous discovery, this enhancement was in bound ferulic acid. It is well-established that bound phenolics released from food matrices during thermal processing and improve the total phenolics of food extracts [20]. Our findings also confirmed this theory, if we observed the patterns of gallic acid (declining of bound gallic acid and enhancement in free gallic acid after processing). In this study, we observed an increase in bound ferulic acid, which further supported our speculation about the formation of phenolics during steaming. Previous studies have pointed out that the formation of phenolics in chestnut might be due to hydrolysis of tannins and lignin [24,29]. Additionally, Gu et al. also stated that starch granules formed during autoclaving, also promote the combination of straight chain starch in chestnut kernels and the formation of hydrogen bonds [30]. These complex reactions might be involving in the release of bound phenolics. 

### 3.3. Effect of Steaming on Cellular Antioxidant Activity (CAA)

In the present study, CAA value of chestnut fruits was up to 2.27 ± 0.10 nmol QE/g FW in free fraction, which was much lower than barley [31] and millet [32]. This might be due to the low concentration of phenolics in chestnut fruit compared to other plants [18], which has been considered to have strong consistency with CAA values [23]. Cellular antioxidant activity was higher in free fractions of chestnut fruits compared to bound fractions in both PBS wash and no PBS wash protocols. These findings were in conformity to the fact that, the amount of phenolic contents and antioxidant activity of free extracts were higher than the bound. Secondly, though steaming had no obvious impact on CAA values before PBS wash in free fractions. However, improvement in CAA after PBS wash was observed, promoting the cellular accessibility of free phytochemicals in chestnut fruit. This might be due to an increase in free phenolics during steaming (Figure 1). In bound fractions, steam processing decreased the cellular antioxidant activity in PBS wash and no PBS wash samples, which might be due to the modifications induced by the steaming process on the chemical structure of bound phenolics that affected the antioxidant properties of this fraction. Nayak et al. reported an increase in the CAA values of potatoes and peas after extrusion [33] and suggested that this process may increase the bio-accessible phytochemicals as well as improve the cellular uptake of phytochemicals. As no further comparison could be made due to the limited literatures, therefore we assumed that steaming may also have the same influence on chestnut fruit. Moreover, our findings revealed that high temperature and proper processing time could improve the cellular accessibility of the free phytochemicals, while the bio-accessibility of bound extracts would be considerably impaired by steaming, but with some damage such as steaming at 121 °C for 20 min.

## 4. Materials and Methods 

### 4.1. Sampling

Chestnut fruit was collected from the Guangdong Academy of Agricultural Sciences, Guangzhou, China. The chestnut shell and inner skin were removed manually. The fresh kernels were processed immediately with seven different treatments: Raw or blank (without treatment), steaming at 100 °C for 10, 20, 30 and 40 min and steaming at 115 and 121 °C for 20 min, respectively. The thermal processing was done by a portable pressure steam sterilizer (DSX-280A, ShenAn Medical, Shanghai, China). The raw as well as processed chestnuts were cooled by liquid nitrogen, separately. After proper grinding, the samples were stored at −80 °C until analyzing.

### 4.2. Extraction 

Free and bound fractions of polyphenolics in chestnut fruit were extracted using the method as reported previously [34]. Briefly, free phytochemicals were extracted with 80% acetone and concentrated by rotary evaporator (Heidolph, Schwabach, Germany) and dissolved with 70% methanol. The residue was further processed to extract bound phytochemicals by NaOH digestion to release phenolics from fibers and carbohydrates and extracted by using ethyl acetate as solvent. The extracted ethyl acetate phase was concentrated by rotary evaporator and dissolved with 70% methanol. Both the free and bound extracts were stored at −80 °C until analyzing.

### 4.3. Determination of Total Phenolics and Flavonoids 

Total phenolic content (TPC) was quantified using the Folin-Ciocalteu assay described previously [34]. Gallic acid was used as a standard to calculate phenolic content, and the data was expressed as mg gallic acid equivalent per 100 g in fresh weight (mg GAE/100 g FW) in triplicates (means ± SD, *n* = 3). Total flavonoids content (TFC) was determined by sodium borohydride/chloranil (SBC) assay as reported previously [35]. Catechin was used as a standard to calculate flavonoid content and the data was expressed as catechin equivalent per 100 g in fresh weight (mg CE/100 g FW) in triplicates (means ± SD, *n* = 3). 

### 4.4. Phenolic Acids and Flavonoids Profiling 

The quantification of phenolic acids and flavonoids in bound and free fractions of chestnut fruit was done by high-performance liquid chromatography with a photodiode array detector (Waters Corp., Milford, MA, USA) [18]. The HPLC conditions were: Mobile phase A with aqueous 0.1% trifluoroacetic acid solution; mobile phase B was methanol; gradient elution (3–5% eluent B in 8 min, then 10% B at 15 min, 20% B at 25 min, 35% B at 33 min, 80% B at 52 min and 5% B at 60 min) with a flow rate of 1.0 mL/min through a Waters Sun FireTM C18 column (250 mm × 4.6 mm, 5 μm) for 60 min; column temperature was set at 30 °C; UV absorbance was set at 280 nm and 324 nm for phenolic acids and flavonoids respectively. Identification of phenolic compounds was achieved by comparison of retention times and recovery rates between standards and samples. The quantification was performed by the standard curves. 

### 4.5. Oxygen Radical Scavenging Capacity (ORAC) Assay

Chemical antioxidant evaluation was carried out using the ORAC assay [36]. In brief, phytochemical extracts were properly diluted with 75 mM phosphate buffer. Then diluted standard (Trolox) or samples with 200 μL fluorescein were added to each well and kept for 20 min at 37 °C. After the additional 20 μL 2,2′-azobis (2-amidinopropane) dihydrochloride (AAPH) solution in each well, the fluorescence intensity was obtained through a microplate reader (Molecular Devices, Sunnyvale, CA, USA) in an interval manner. Data were reported as micromole Trolox equivalent per gram in fresh weight (μmol TE/g FW). All the data was repeated three times and expressed as mean ± SD (*n* = 3).

### 4.6. Cellular Antioxidant Activity (CAA) Assay

The CAA assay was conducted as described previously [23]. Human live cancer cell line (HepG2, ATCC HB-8065) was used as the cellular model in this assay; quercetin was used as a standard to calculate the cellular antioxidant activity value. HepG2 cells were seeded at a density of 6 × 10^4^ cells/well on a 96-well microplate for antioxidant activity analysis. Dichlorofluorescin diacetate (DCFH-DA) was used as fluorescence probe and AAPH was used as a free radical donor. With and without PBS wash treatments were used in this assay. Fluorescence intensity was measured at excitation of 485 nm and emission of 535 nm for a dynamic analysis by Microplate Reader (Molecular Devices, Sunnyvale, CA, USA). The CAA value was calculated from the integrated area under the fluorescence versus time curve, and the results were expressed as nanomole of quercetin equivalent (QE) per gram in fresh weight (nmol QE/g FW). 

### 4.7. Statistics Analysis 

All data were reported as mean ± SD (*n* = 3). Data were analyzed among groups using one-way analysis of variance (ANOVA) and Tuckey’s multiple comparison post-test. Significance analysis and Pearson correlation were calculated using SPSS 13.0 (SPASS Inc, Chicago, IL, USA), whereas, SigmaPlot 11.0 (Systat Software, Inc, Chicago, IL, USA) was used to present data in graphical format.

## 5. Conclusions

Our findings revealed that steaming time and temperature had a varied influence on these indexes, and longer processing time and higher temperature tended to improve the levels of phenolics, flavonoids as well as total and cellular antioxidant activities in free fractions of chestnut. A detailed study on the antioxidant activities of individual phenolic compounds as well as their changes could be helpful for an in depth understanding of phenolics formation during thermal processing.

## Figures and Tables

**Figure 1 molecules-24-00703-f001:**
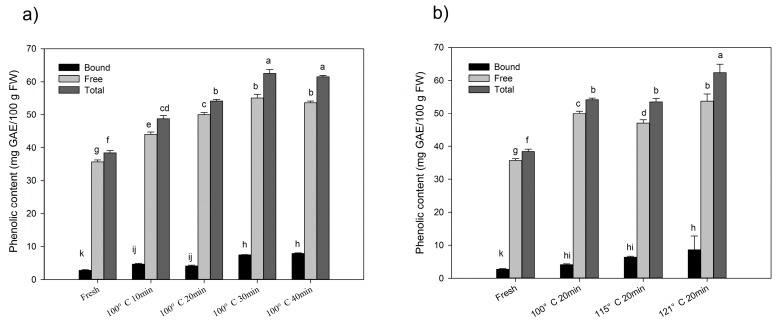
(**a**) Changes of phenolic contents with time-kinetics manner at 100 °C; (**b**) changes of phenolic contents with a thermal gradient manner at 20 min steaming processing. Significant differences (*p* < 0.05) exist among those bars with different letters by Tuckey analysis.

**Figure 2 molecules-24-00703-f002:**
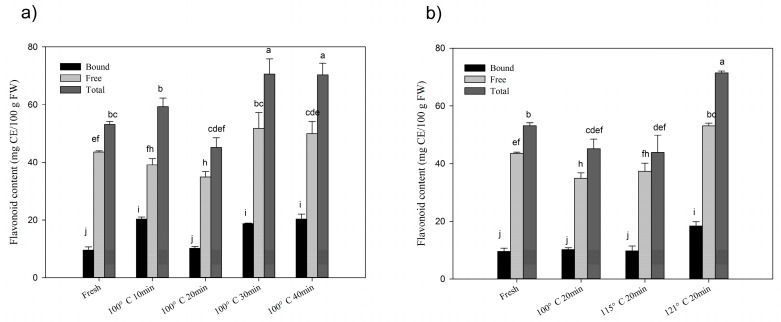
(**a**) Changes of flavonoid contents with time-kinetics manner at 100 °C; (**b**) changes in flavonoid contents with thermal gradient manner at 20 min steaming processing. Significant differences (*p* < 0.05) exist amongst those bars with different letters by Tuckey analysis.

**Figure 3 molecules-24-00703-f003:**
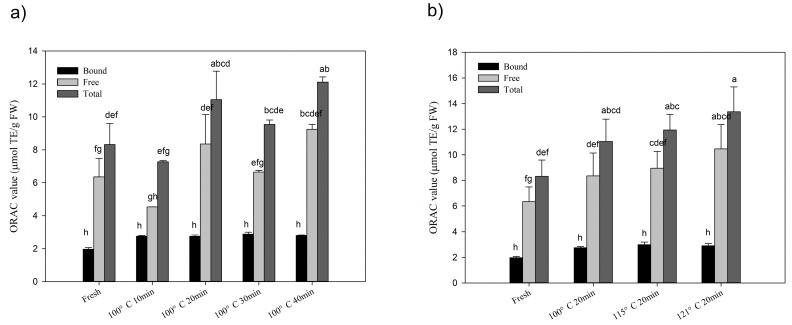
(**a**) Changes of antioxidant ability with time-kinetics manner at 100 °C; (**b**) changes of antioxidant ability with thermal gradient manner at 20 min steaming processing. Significant differences (*p* < 0.05) exist among those bars with different letters by Tuckey analysis.

**Table 1 molecules-24-00703-t001:** Changes of phytochemical profiles in chestnut fruits after thermal processing.

Compound	Conjugation	Fresh	100 °C 10 min	100 °C 20 min	100 °C 30 min	100 °C 40 min	115 °C 20 min	121 °C 20 min
Ferulic acid	free	2.98 ± 0.01 ^a^	2.91 ± 0.01 ^b^	2.93 ± 0.01 ^b^	2.83 ± 0.01 ^c^	2.83 ± 0.01 ^c^	2.92 ± 0.03 ^b^	2.73 ± 0.01 ^d^
	bound	2.92 ± 0.43 ^e^	3.57 ± 0.01 ^bc^	3.24 ± 0.02 ^c,d^	4.16 ± 0.05 ^a^	4.10 ± 0.01 ^a^	3.88 ± 0.01 ^a,b^	3.58 ± 0.01 ^b,c^
Chlorogenic acid	free	18.28 ± 0.14 ^d^	15.16 ± 0.13 ^e^	18.80 ± 0.12 ^c^	18.81 ± 0.08 ^c^	22.42 ± 0.01 ^a^	21.13 ± 0.28 ^b^	21.17 ± 0.16 ^b^
	bound	ND	ND	ND	ND	ND	ND	ND
Gallic acid	free	9.76 ± 0.30 ^d^	9.61 ± 0.09 ^d^	12.93 ± 0.06 ^a^	11.72 ± 0.03 ^c^	12.54 ± 0.02 ^b^	4.99 ± 0.11 ^e^	5.32 ± 0.01 ^e^
	bound	5.55 ± 0.01 ^a^	2.86 ± 0.01 ^e^	2.92 ± 0.01 ^d^	2.85 ± 0.01 ^e^	2.86 ± 0.01 ^e^	3.94 ± 0.01 ^c^	5.47 ± 0.01 ^b^
Vanillic acid	free	6.38 ± 0.14 ^b^	6.47 ± 0.01 ^a,b^	6.38 ± 0.01 ^b^	6.54 ± 0.01 ^a,b^	6.65 ± 0.01 ^a^	6.47 ± 0.11 ^a,b^	6.52 ± 0.04 ^a,b^
	bound	ND	ND	ND	ND	ND	ND	ND
Syringate	free	3.95 ± 0.06 ^a^	2.74 ± 0.02 ^e^	3.43 ± 0.02 ^b^	3.26 ± 0.09 ^c^	2.97 ± 0.05 ^d^	3.19 ± 0.07 ^c^	2.72 ± 0.01 ^e^
	bound	ND	ND	ND	ND	ND	ND	ND
Quercetin	free	1.70 ± 0.11 ^a^	0.40 ± 0.01 ^d^	0.53 ± 0.01 ^c,d^	0.59 ± 0.04 ^c,d^	0.37 ± 0.01 ^d^	1.39 ± 0.19 ^b^	0.63 ± 0.01 ^c^
	bound	0.90 ± 0.01 ^c^	1.62 ± 0.01 ^a^	1.13 ± 0.10 ^b^	0.75 ± 0.06 ^d^	0.99 ± 0.05 ^b,c^	0.84 ± 0.01 ^c,d^	1.58 ± 0.07 ^a^
*p*-Coumaric acid	free	2.38 ± 0.01 ^c,d^	2.39 ± 0.01 ^c^	2.39 ± 0.01 ^c^	2.39 ± 0.01 ^c^	2.37± 0.01 ^d^	2.43± 0.01 ^b^	2.52± 0.01 ^a^
	bound	2.41 ± 0.01 ^d,e^	2.77 ± 0.01 ^a^	2.42 ± 0.01 ^c,d^	2.43 ± 0.01 ^c^	2.41± 0.01 ^d,e^	2.43± 0.01 ^c^	2.46 ± 0.01 ^b^

Notes: Unit, mg/100 g FW. Turkey tests were carried out in each row and significant differences (*p* < 0.05) exist among those with different letters. ND means not detected.

**Table 2 molecules-24-00703-t002:** Changes of cellular antioxidant activity (CAA) values in chestnut fruits after thermal processing.

Sample Group	CAA Value (nmol QE/g FW)				Cellular Uptake (%)	
	No-PBS Wash		PBS Wash			
	Free	Bound	Free	Bound	Free	Bound
Fresh	1.69 ± 0.51 ^b^	0.19 ± 0.01 ^a,b^	0.24 ± 0.01 ^d,e^	0.17 ± 0.04 ^a^	14.3% ^d^	88.1% ^a^
100 °C 10 min	1.61 ± 0.16 ^b^	ND	0.20 ± 0.05 ^e^	ND	12.3% ^d^	ND
100 °C 20 min	1.52 ± 0.07 ^b^	0.19 ± 0.01 ^a,b^	0.28 ± 0.01 ^c,d,e^	ND	18.6% ^c^	ND
100 °C 30 min	1.69 ± 0.07 ^b^	ND	0.45 ± 0.06 ^a^	ND	26.3% ^a^	ND
100 °C 40 min	2.27 ± 0.10 ^a^	0.11 ± 0.04 ^c^	0.31 ± 0.01 ^b,c^	ND	13.4% ^d^	ND
115 °C 20 min	1.81 ± 0.05 ^b^	0.24 ± 0.01 ^a^	0.34 ± 0.01 ^b^	0.06 ± 0.02 ^b^	18.7% ^c^	26.8% ^c^
121 °C 20 min	1.61 ± 0.02 ^b^	0.14 ± 0.04 ^b,c^	0.33 ± 0.03 ^b,c^	0.07 ± 0.02 ^b^	20.3% ^b^	48.2% ^b^

Note: Turkey tests were carried out in each column and significant differences (*p* < 0.05) exist among those with different letters. ND means not detected. Cellular uptake = CAA values of PBS wash/CAA values of no PBS wash in percentage (100%).

**Table 3 molecules-24-00703-t003:** Pearson correlation coefficient among phenolic profiles, contents and total and cellular antioxidant activities.

Correlation	ORAC	CAA No-PBS Wash	CAA PBS Wash	TPC	TFC	Thermal Processes
ORAC	-	0.40	0.23	0.59 *	0.17	0.78 **
CAA No-PBS wash	-	-	0.12	0.16	0.11	0.21
CAA PBS wash	-	-	-	0.10	0.16	−0.44 *
TPC	-	-	-	-	0.61 **	0.56 *
TFC	-	-	-	-	-	0.18
TFA	0.24	0.36	0.07	0.65 **	0.42	0.41
TCA	0.82 **	0.70 **	0.42	0.55 **	0.25	0.66 **
TGA	−0.30	0.05	−0.15	−0.13	0.06	−0.84 **
TVA	0.29	0.35	0.06	0.63 **	0.68 **	0.40
TS	−0.29	0.05	0.39	−0.62 **	−0.51 *	−0.57 *
TQ	−0.19	−0.14	0.17	−0.69 **	−0.39	−0.01
TPA	−0.29	−0.49 *	−0.55 *	−0.08	0.12	0.19

Note: The phenolic profiles are TPC (total phenolics), TFC (total flavonoids), TFA (total ferulic acid), TCA (total chlorogenic acid), TGA (total gallic acid), TVA (total vanillic acid), TS (total syringate), TQ (total quercetin), TPA (total *p*-coumaric acid). Total = free + bound * and ** mean correlation is significant at the 0.05 and 0.01 level respectively (2-tailed).

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
