# Peer review of "Effect of Steaming Processing on Phenolic Profiles and Cellular Antioxidant Activities of Castanea mollissima"

_molecules, 2019, doi:10.3390/molecules24040703_

Round 1
Reviewer 1 Report
The results reported in the manuscript seem interesting and worth to be published, although editing of English language and style is required in some parts of the manuscript (e.g. lines 23-24, 25-26, 52-53, line 145 ecc.) In particular, I would suggest to considerably revise and rephrase the discussion section, because in my opinion it is too complex and repetitive. In addition, some other points should be addressed:
1) Phenol contents are expressed as mg/g FW for both the fresh and processed chestnuts: how did authors take into account the weight loss likely occurring during steaming (due e.g. to water loss)?
2) Lines 188-120: authors should replace “1.5 fold” with “1.2 fold” since the starting value is 18.28 and not 15.16.
3) Lines 127-129: authors stated that quercetin behaves like gallic acid during thermal processing; indeed no increase in quercetin levels was observed, in contrast to gallic acid, so please rephrase the sentence.
4) Table 2: I would suggest authors to perform statistical analysis also for the cellular uptake levels.
5) Line 206-213: I was not able to catch the meaning of the term “synthesized”… Why did authors rule out the possibility of a release of chlorogenic acid during steaming? Authors should better discuss about this point.
6) Lines 248-250: Authors attributed the lower cellular antioxidant activity of the bound fraction to “a decline in bound phytochemicals”. In my opinion it is not the amount of bound phytochemicals (if I well understood the assay was run on the bound extract as a powder) but rather the modifications induced by the steaming process on the chemical structure of bound polyphenols that affected the antioxidant properties of this fraction.
7) Line 261 and lines 280-281: I was not able to find these correlations in Table 3.
8) Table 3: a comment should be added on the correlation coefficients found between TPA and CAA No-PBS wash and CAA PBS wash.
9) Line 296: which types of bonds are affected by the NaOH treatment?
10) Line 344: “formation” should be replaced with “release”.
Author Response
Reviewer 1
The results reported in the manuscript seem interesting and worth to be published, although editing of English language and style is required in some parts of the manuscript (e.g. lines 23-24, 25-26, 52-53, line 145 ecc.) In particular, I would suggest to considerably revise and rephrase the discussion section, because in my opinion it is too complex and repetitive. In addition, some other points should be addressed:
1) Phenol contents are expressed as mg/g FW for both the fresh and processed chestnuts: how did authors take into account the weight loss likely occurring during steaming (due e.g. to water loss)?
Response: As we used steaming treatments in this study for chestnut processing, the moisture was consisted about 100% in different treatments and the weight of chestnut fruits was little changed during thermal processing. Therefore, fresh weight was used to expressed all the data in this study.
2) Lines 188-120: authors should replace “1.5 fold” with “1.2 fold” since the starting value is 18.28 and not 15.16.
Response: I am sorry to make this mistake. It was corrected.
3) Lines 127-129: authors stated that quercetin behaves like gallic acid during thermal processing; indeed no increase in quercetin levels was observed, in contrast to gallic acid, so please rephrase the sentence.
Response: Thank you very much for your comments. I am sorry to make confusion for this sentence. It was revised.
4) Table 2: I would suggest authors to perform statistical analysis also for the cellular uptake levels.
Response: Thank you very much for your suggestion. It was added.
5) Line 206-213: I was not able to catch the meaning of the term “synthesized”… Why did authors rule out the possibility of a release of chlorogenic acid during steaming? Authors should better discuss about this point.
Response: I am sorry to make confusion in this part. It was revised in the manuscript.
6) Lines 248-250: Authors attributed the lower cellular antioxidant activity of the bound fraction to “a decline in bound phytochemicals”. In my opinion it is not the amount of bound phytochemicals (if I well understood the assay was run on the bound extract as a powder) but rather the modifications induced by the steaming process on the chemical structure of bound polyphenols that affected the antioxidant properties of this fraction.
Response: Thank you very much for your comments. I am sorry to confuse this sentence. It was revised.
7) Line 261 and lines 280-281: I was not able to find these correlations in Table 3.
Response: Line 261, as table 3 showed, total flavonoids were correlated with total phenolics as 0.61** at p < 0.01. Line 280-281, as table 3 showed, chlorogenic acid was correlated with CAA No-PBS wash as 0.70** at p < 0.01.
8) Table 3: a comment should be added on the correlation coefficients found between TPA and CAA No-PBS wash and CAA PBS wash.
Response: Thank you very much for your suggestion. It was added.
9) Line 296: which types of bonds are affected by the NaOH treatment?
Response: NaOH treatment was aimed to release bound phenolics from fibers and carbohydrates.
10) Line 344: “formation” should be replaced with “release”.
Response: It was changed.
Reviewer 2 Report
General comments
This is a descriptive manuscript concerning the effect of steaming processing on the extraction of free and bound phenolics, flavonoids as well and antioxidant activities in chestnuts. Steaming processing is carried out at different times and temperatures. Trends are Ok, interesting and with possible applications, as it is shown that the increase in time or temperature usually increases the amount of phenolics extracted.
1) However, some punctual data are difficult to justify. See for instance Figure 2A, extraction at 100ºC during 20 min in comparison to 10 min and 30 min. Results showed at Table 1 concerning the HPLC determination of specific phenols do not corroborate the diminished extraction after 20 min (i.e. see gallic or chlorogenic acids, the most abundant ones). At Figure 3a, concerning antioxidant activity, the punctual odd data is obtained at 100ºC, 30 min (lower than 20 and 40 min). In discussion, these odd data are mentioned and justified in a confused form. I recommend the emphasis in the trends rather than these odd data. In spite of the p value, I think that the statistic is not totally convinced and the logical science throws some doubts about that.
Discussion is long and a little bit tedious. It should be shortened, going directly to the main points, avoiding the descriptive reiteration of results. The possible reasons of the increase in free and bound phenolics compounds should be briefly discussed. It is very unlikely to speculate about net synthesis of phenolics compounds (i.e. line 209 and others). It is much more likely the increase due to hydrolysis of conjugated compounds in the organic matrix of chestnuts, as mentioned at lines 222-223 (“Previous studies have pointed out that the formation of phenolics in chestnut might be due to hydrolysis of tannins and lignin[23, 28]”).
2) The differences of this study with the ref. 18 should be emphasized. What is really new in this new manuscript? For instance….
Lines 64-65: Such as in our previous study, compared to boiling and microwaving, the chestnut fruit processed by steaming had highest contents of total phenolics and flavonoids [18].
Or line 73: our previous study has indicated that steaming could dramatically increase both the bound and free phenolics in chestnut fruit[18].
3) Line 66: Define clearly the term "bound phenolics". Are polyphenols "bound phenolics"?. What about glycosides?. Are glycosides hydrolyzed by thermal treatment to release the aglicone moiety from the carbohydrate?. In principle, steaming and other thermal treatments could also increase the amount of bound phenolics by chemical reactions, as the oxidations of o-diphenols to o-quinones give place to polymers and pseudo-melanin material causing the browning of the vegetal tissue.
4) Concerning methods and the determination of cellular antioxidant activity, the reasons and expected differences of two alternatives, PBS wash and no PBS wash, should be justified.
Minor points
Line 321: Something missing about AAPH regular name? Is that [2,2’-azobis(2-
methylpropionamidine) dihydrochloride?
Lines 329- 330: Some abbreviations are difficult to follow DCFH-DA, ABAP. Please, use the complete name or a list of abbreviations at the end of the manuscript
Conclusions
Lines 349-351: Of course, but this is not a conclusion. Author should be more precise, about free and bound phenols. On the other hand, the formation of novel phenols mentioned at line 344 should be clarified. This formation is a little bit surprising, as mentioned earlier. What that is means?
Author Response
Reviewer 2
General comments
This is a descriptive manuscript concerning the effect of steaming processing on the extraction of free and bound phenolics, flavonoids as well and antioxidant activities in chestnuts. Steaming processing is carried out at different times and temperatures. Trends are Ok, interesting and with possible applications, as it is shown that the increase in time or temperature usually increases the amount of phenolics extracted.
1) However, some punctual data are difficult to justify. See for instance Figure 2A, extraction at 100ºC during 20 min in comparison to 10 min and 30 min. Results showed at Table 1 concerning the HPLC determination of specific phenols do not corroborate the diminished extraction after 20 min (i.e. see gallic or chlorogenic acids, the most abundant ones). At Figure 3a, concerning antioxidant activity, the punctual odd data is obtained at 100ºC, 30 min (lower than 20 and 40 min). In discussion, these odd data are mentioned and justified in a confused form. I recommend the emphasis in the trends rather than these odd data. In spite of the p value, I think that the statistic is not totally convinced and the logical science throws some doubts about that.
Discussion is long and a little bit tedious. It should be shortened, going directly to the main points, avoiding the descriptive reiteration of results. The possible reasons of the increase in free and bound phenolics compounds should be briefly discussed. It is very unlikely to speculate about net synthesis of phenolics compounds (i.e. line 209 and others). It is much more likely the increase due to hydrolysis of conjugated compounds in the organic matrix of chestnuts, as mentioned at lines 222-223 (“Previous studies have pointed out that the formation of phenolics in chestnut might be due to hydrolysis of tannins and lignin[23, 28]”).
Response: Thank you very much for your comments and suggestion. Total phenolics changed with the increased time and temperature during thermal processing, the results were accordance to the phenolic profiles changes by HPLC. While, total flavonoids declined at 100ºC for 20 min compared with other treatments, the results were not accordance to total phenolics and phenolic profiles. We speculated that some unknown flavonoid compounds affected this fluctuation for total flavonoids, not detected in our study. The HPLC data are most phenolic acids, only one flavonoid compound (quercetin) were detected in this study. That’s why the figure 2a did not accordance to table 1of HPLC data. The following research will focus on flavonoid composition and profiles during different processes in chestnut.
For discussion part, we have revised the manuscript carefully according to your suggestion.
2) The differences of this study with the ref. 18 should be emphasized. What is really new in this new manuscript? For instance….
Lines 64-65: Such as in our previous study, compared to boiling and microwaving, the chestnut fruit processed by steaming had highest contents of total phenolics and flavonoids [18].
Or line 73: our previous study has indicated that steaming could dramatically increase both the bound and free phenolics in chestnut fruit[18].
Response: Thank you very much for your suggestion. The reference 18 has been discussed and compared with our present study in the manuscript.
3) Line 66: Define clearly the term "bound phenolics". Are polyphenols "bound phenolics"?. What about glycosides?. Are glycosides hydrolyzed by thermal treatment to release the aglicone moiety from the carbohydrate?. In principle, steaming and other thermal treatments could also increase the amount of bound phenolics by chemical reactions, as the oxidations of o-diphenols to o-quinones give place to polymers and pseudo-melanin material causing the browning of the vegetal tissue.
Response: Thank you very much for your suggestion. It was revised and described in introduction part.
4) Concerning methods and the determination of cellular antioxidant activity, the reasons and expected differences of two alternatives, PBS wash and no PBS wash, should be justified.
Response: Thank you very much for your suggestion. It was revised and described in the manuscript.
Minor points
Line 321: Something missing about AAPH regular name? Is that [2,2’-azobis(2-methylpropionamidine) dihydrochloride?
Response: It was added.
Lines 329- 330: Some abbreviations are difficult to follow DCFH-DA, ABAP. Please, use the complete name or a list of abbreviations at the end of the manuscript
Response: It was added and revised.
Conclusions
Lines 349-351: Of course, but this is not a conclusion. Author should be more precise, about free and bound phenols. On the other hand, the formation of novel phenols mentioned at line 344 should be clarified. This formation is a little bit surprising, as mentioned earlier. What that is means?
Response: Thank you very much for your suggestion. It was revised.
Reviewer 3 Report
The document focus on the effect of steaming processing on phenolic profiles and cellular antioxidant activities of Castanea mollissima, although scarce information related to the extraction methodology and biological activity of this product is presented in the literature more information could be provided.
In the abstract more information about the compounds presented in the phenolic profile must be added.
In legend of figures 1 and 2 the test to obtain significant differences must be added. (example Tuckey or LCD)
Why did the authors use just one method to analyzed antioxidant activities if it has been demonstrated that at least 2 methods must be used to give consistent information?
A mathematical model could be used to described a correlation between factors (temperature and time) to phenolic compounds increment
In line 190 the authors suggest that high energy is required for disruption of food matrixes, values of the energy must be provided using experimental or theoretical data as well as reported information.
In line 209 authors mentioned that novel phenolic compounds were synthesized during steaming, nevertheless according to the data obtained by HPLC presented here in there are not new compounds described in processed samples compare to the fresh ones. A better discussion is needed.
In Materials and Methods section the methodology for the quantification of the compounds must be detailed.
Author Response
Reviewer 3
The document focus on the effect of steaming processing on phenolic profiles and cellular antioxidant activities of Castanea mollissima, although scarce information related to the extraction methodology and biological activity of this product is presented in the literature more information could be provided.
Response: The methods related to the extraction methodology and biological activity in this study have been cited from their original articles as references in the manuscript. We have not modified any procedures in this study. That’s why we omitted the operational details in the manuscript. More details can be found in the original articles as cited references.
In the abstract more information about the compounds presented in the phenolic profile must be added.
Response: Thank you very much for your suggestion. It was revised and added in the abstract.
In legend of figures 1 and 2 the test to obtain significant differences must be added. (example Tuckey or LCD)
Response: It was added in figure 1 and figure 2.
Why did the authors use just one method to analyzed antioxidant activities if it has been demonstrated that at least 2 methods must be used to give consistent information?
Response: Two methods of antioxidant activity including ORAC and CAA were used in this study. ORAC assay is a method of measuring antioxidant capacities in biological samples in vitro and was popular used to evaluate foods antioxidant activity as AOAC method. CAA assay simulates some of the cellular processes including bio-accessibility, cell uptake, metabolism, and distribution of bioactive compounds to predict the antioxidant behavior in biological systems. In CAA assay with PBS wash and No-PBS wash treatments, the cellular accessibility of extracts could be evaluated, which is helpful to predict the antioxidant ability of plant extracts in vivo. That’s why we chose these two assays in this study.
A mathematical model could be used to described a correlation between factors (temperature and time) to phenolic compounds increment
Response: It was revised.
In line 190 the authors suggest that high energy is required for disruption of food matrixes, values of the energy must be provided using experimental or theoretical data as well as reported information.
Response: It was revised.
In line 209 authors mentioned that novel phenolic compounds were synthesized during steaming, nevertheless according to the data obtained by HPLC presented here in there are not new compounds described in processed samples compare to the fresh ones. A better discussion is needed.
Response: Thank you very much for your comments. I am sorry to make confusion in this part. It was revised in the manuscript.
In Materials and Methods section the methodology for the quantification of the compounds must be detailed.
Response: Thank you very much for your suggestion. It was added.
Round 2
Reviewer 2 Report
Authors address most of the points arisen in the previous report, and the manuscript has been modified and improved according the reply letter.